# Influence of Pig Genetic Line and Salt Reduction on Peptide Production and Bioactivity of Dry-Cured Hams

**DOI:** 10.3390/foods12051022

**Published:** 2023-02-28

**Authors:** Beatriz Muñoz-Rosique, Noelia Hernández-Correas, Adela Abellán, Estefanía Bueno, Rafael Gómez, Luis Tejada

**Affiliations:** 1Departamento de Calidad, AromaIbérica Serrana, S.L. Ctra. Fuente Álamo, Km 17.4, 30332 Murcia, Spain; 2Departamento de Tecnología de la Alimentación y Nutrición, Universidad Católica de Murcia, Campus de los Jerónimos, 30107 Murcia, Spain; 3Departamento de Bromatología y Tecnología de los Alimentos, Universidad de Córdoba, Campus de Rabanales, Edificio Darwin, 14014 Córdoba, Spain

**Keywords:** bioactive peptide, salt reduction, proteolysis, deboned ham, Iberian ham, antioxidant, angiotensin-I converting enzyme (ACE)

## Abstract

Ham (Jamón) is a product of great value in Spanish gastronomy, although experts have recommended reducing its consumption due to its high salt content and its relationship with cardio-vascular diseases due to the increase in blood pressure it may cause. Therefore, the objective of this study was to evaluate how the reduction of salt content and the pig genetic line influence bioactivity in boneless hams. For this purpose, 54 hams were studied, 18 boneless Iberian hams (RIB), 18 boneless white hams from commercial cross-bred pigs (RWC), and 18 salted and traditionally processed Iberian hams (TIB) to check if the pig genetic line (RIB vs. RWC) or the processing (RIB vs. TIB) affect the peptide production and bioactivity of the hams. The pig genetic line significantly affected the activity of ACE-I and DPPH, with RWC having the highest ACE-I activity and RIB having the highest antioxidative activity. This coincides with the results obtained in the identification of the peptides and the bioactivity analysis performed. Salt reduction positively affected the different hams, influencing their proteolysis and increasing their bioactivity in traditionally cured hams.

## 1. Introduction

In recent years, due to the increasing concern of the population for health and nutrition, there is a tendency to investigate and demonstrate the added value and nutritional properties of certain foods. Therefore, the term “Functional Foods” is becoming increasingly prevalent and these foods could prevent the appearance or improve symptoms of certain chronic diseases [1].

Meat products have also been studied for this bifunctionality [2], and numerous proposals have been reformulated to reduce or even eliminate certain components such as salt [3] or fat, seeking to improve the nutritional profile of the food. One of the most important challenges in the food industry is the development of functional foods that maintain the organoleptic characteristics necessary to meet consumer demands, especially regarding flavor and texture [4,5]

Bioactive peptides are a group of biological molecules normally buried in the structure of parent proteins that become active after the cleavage of the proteins. During the ham curing process, the action of endopeptidases (capthesins and calpains) and exoproteases is crucial for the formation of peptides. Capthepsins and calpeins begin their action during the cold phase of the dryer (post-salting stage), but the increase in salt concentration, together with drying, causes their activity to decrease. However, exoprotease activity is highly favored when the temperature is increased in the dryer above 25 °C (temperature-dependent activity) and is maintained until the most advanced stages of the process. Intense proteolysis is thus triggered, leading to the hydrolysis of proteins and the release of non-protein nitrogenous compounds (NPNs). Due to this process, free amino acids and small peptides accumulate. These would comprise 2–20 amino acids (AA) with a molecular mass less than 6000 Da [6].

The natural generation of these bioactive peptides is a consequence of the intense proteolysis of muscle peptidases produced during the processing of cured ham [4,7]. However, there is still little information on the amount of these peptides in the final product. Due to the large number of bioactive peptides, the low abundance of each one and their presence within a complex matrix such as dry-cured ham makes their extraction and analysis difficult [7,8,9]. Peptides modify the texture of the cured meat and influence the aroma at the end of the processing [4,10,11].

Numerous studies have described the beneficial properties that these peptides may have on health. The influence they may have on certain diseases of chronic evolution has been evaluated, exerting antioxidant activity [12,13,14], antihypertensive activity [15,16], immunomodulatory [17], appetite regulator activity [18], or antidiabetogenic activity [19]; with good results and evidence for the ability to inhibit the angiotensin-I converting enzyme (ACE-I) [15,20,21].

The direct or indirectly prescription of these peptides, which are naturally derived from food, could treat certain diseases and thus circumvent adverse effects secondary to the use of artificial drugs. Other authors described strategies to increase the consumption of meat products in meals, to add precursor proteins of these functional peptides to certain meals or to directly add the peptides, after studying their encapsulation [22].

Salt plays a fundamental role during the salting period of cured ham. Water retention capacity, texture, flavor, pH, and proteolysis are parameters directly influenced by salt content [23,24]. Despite being one parameter that defines the quality of cured ham, and being physiologically necessary for the proper function of an organism [25], the current excessive consumption of high salt products, such us ultra-processed foods has caused current human populations to exceed the necessary daily intake of this mineral, causing hypertension, directly related to cardiovascular diseases and other diseases of risk [26,27]. Therefore, numerous studies are being conducted on various foods to reduce their salt content.

In the case of ham, salt reduction directly influences proteolysis, which is greaterwhen the salt content is lower, and which, together with water activity, will result in a softer texture of the ham, which, as a consequence, will alter the quality of the final product [28]. The increase in proteolysis leads to protein degradation, resulting in the generation of free AA and peptides through the proteolytic action of endopeptidases that can have biological activity, which can counteract and/or prevent diseases [29].

Scant research has been performed on meat-derived peptides, and the degree to which salt reduction has influenced the generation of peptides or the generation of bioactive peptides has not been investigated.

Numerous authors have identified peptides with antihypertensive and antioxidant activity in different types of cured ham [29] that show in vitro ACE inhibitory activity, most important in the peptides Ala-Ala-Pro-Leu-Ala-Pro and Ile-Ala-Gly-Arg-Pro [30]. Some of the identified peptides showed multifunctional activity, i.e., some showed antioxidant or anti-inflammatory activity besides antihypertensive activity [29].

Some trials have been conducted with peptides found in ham to test whether they had bioactivity. A study conducted in hypertensive rats showed a significant reduction in systolic pressure after administering a peptide with antihypertensive activity (ACE inhibitor) eight hours earlier [31]. Furthermore, this multifunctional activity had been recognized from bioactive peptides in other foods [32], most commonly generated from hydrophobic waste [33].

Despite advances made in this field, the effect of salt reduction in ham on peptide generation and bioactivity has not been studied. Furthermore, no studies have been found comparing the peptides present in Iberian ham with those present in white ham, and no studies have attempted to identify peptides in boneless hams. Therefore, the objective of this study was to evaluate the influence of the reduction of salt content and pig genetic line on peptide production and bioactivity in boneless cured hams.

## 2. Materials and Methods

For this study, 54 hams were selected from different genetic lines of Iberian pigs (thirty-six hams with a racial percentage of 50% Iberian and 50% Duroc) and white pigs (eighteen hams from crossbreeding Landrace x Large White or Hampshire).

The hams from the Iberian pork genetic line were distributed in six batches of six hams. Half were processed on bone traditionally (18 TIB hams) and were the control batches. The other half of the Iberian hams (18 RIB hams) and the three batches of white hams (18 RWC hams) were freshly deboned and subjected to the new process developed to achieve salt reduction (RIB and RWC, respectively). To evaluate the effect of processing (deboning and salt reduction), TIB and RIB hams were compared, so only salt reduction in the Iberian hams was compared. To study the effect of the genetic line, RIB and RWC hams were compared.

To compare hams from different genetic lines, the percentage of loss was taken as a reference. During the processing phases, each sample was weighed in triplicate to determine the percentage weight loss of each ham respect to the initial fresh weight of each piece. We considered the optimal curing moment, or the final product, when the ham reached 38% of weight loss. The experimental design is shown in Figure 1.

Fresh hams were deboned and salted using sea salt and nitrifying salts. These hams were kept in a cold room at 3 °C for an established period (0.8 days per kilogram of ham weight). The hams were then removed and washed with water, following the normal curing process [34]. The next stage (rest or post-salting) was conducted at 3 °C, gradually increasing this temperature to 6 °C until the percentage of weight loss of the hams rose to 18%. When this phase ended, the temperature was increased to 28 °C, reaching its completion when the hams reached a loss of 38%. Once the hams were cured, 18 hams were selected (six TIB hams, six RIB hams and six RWC hams). The samples were taken during the drying stage (33% weight loss) and final product stage (38% weight loss). All samples were taken when the ham reached the required percentage of weight loss. To avoid damage to the piece during the sampling process, for all the analysis all samples were taken from the femoral muscle (biceps) using a stainless steel cylinder with a diameter of 2 cm. After, samples were kept refrigerated until their analysis.

### 2.1. Non-Protein Nitrogenous Compounds

To prepare the extracts, 2 g of sample were weighed in an Erlenmeyer flask, after which 30 mL of distilled water were added, and agitated for 15–20 min in a magnetic stirrer. Then, 15 mL of 20% trichloroacetic acid was added and shaken for 10 min. The content of the Erlenmeyer was filtered using a funnel and filter paper in a 50 mL volumetric flask. After filtration, the flask was filled with distilled water [35]. Finally, 10 mL of the extract were used for the Kjeldahl method [36].

### 2.2. Antioxidant Activity

The determination of antioxidant activity was conducted following the method of Bersuder et al. (1998) [37] with minor variations. First, a standard was created with the TROLOX reagent (6-hydroxy-2,5,7,8-tetramethylchroman-2-carboxylic acid), which is an analog of vitamin E with antioxidant capacity, used to react with the DPPH radical (2,2-diphenyl-1-picrylhydrazyl). The stock solution of TROLOX (2 Mm) was prepared by weighing 12.5 mg of TROLOX and diluting it in 25 mL of ethanol. A second 0.1 mM stock solution of 10 mL was then prepared from the first stock solution of 2 Mm. This second stock solution was prepared with 0.5 mL of the first stock solution and 9.5 mL of ethanol. The TROLOX concentrations of the straight standard were 5, 10, 15, 20, 40, and 50 µM. Two milliliters of each of the concentrations were prepared from the second 0.1 mM stock solution to prepare the standard line, obtaining the following linear equation: y = 1.01x + 10.224 (R^2^ = 0.9988).

The antioxidant activity of the samples was then determined using a 0.02% (*w*/*v*) solution of the DPPH radical in ethanol. In Eppendorf tubes, 500 µL of ethanol, 500 µL of sample, and 125 µL of 0.02% (*w*/*v*) DPPH solution were added. For the blank, 500 µL of ethanol, 500 µL of water, and 125 µL of DPPH were used. Samples were incubated for 1 h in the dark at room temperature. The samples were then centrifuged at 10,000× *g* for 2 min and their absorbance was measured at 517 nm.
% DPPH RSA = (Abs Blank − Abs Sample)/Abs Blank × 100
where, % DPPH RSA is % DPPH radical scavenging activity. Abs blank is the absorbance of DPPH with water instead of hydrolysate and Abs sample is the absorbance of DPPH radical in the presence of hydrolysate. The analyses for the different hydrolysates were performed three times.

To calculate the IC_50_ (concentration of our sample that inhibits 50% of the DPPH radical) of our sample, we use the equation of the TROLOX standard we performed previously.

### 2.3. Angiotensin-I-Converting Enzyme Inhibitory Activity

The ACE-I activity of the hydrolysates was conducted according to the spectro-photometric method of Cushman and Cheung (1971) [38], modified by Miguel et al. (2004) [39].

For this purpose, 40 µL of each hydrolysate was incubated at 37 °C with 100 µL of 5 mM HHL dissolved in 0.1 M borate buffer and 0.3 M NaCl (pH 8.3). Next, 2 mU of ECA was added to the substrate. Thirty minutes later, 150 µL of 1 M HCl was added. The formed hippuric acid was recovered with 1000 µL ethyl acetate and centrifuged 10 min at 4000× *g* and the organic phase (800 µL) was collected. The ethyl acetate was removed by bringing the temperature to 95 °C. The resulting hippuric acid was resuspended in 1000 µL of distilled water and the absorbance was measured at 228 nm. ACE inhibitory activity was determined by the following equation
(1)$$ACE\;inhibitory\;activity\;(\%)=\left(\frac{\left(A_{\mathrm{control}}-A_{\mathrm{blank}})-(A_{\mathrm{sample}}-A_{\mathrm{blank}}\right)}{\left(A_{\mathrm{control}}-A_{\mathrm{blank}}\right)}\right)\times 100$$
where,

A_control_ is the measure of hippuric acid produced by the action of uninhibited ACE, A_sample_ is the measure of hippuric acid produced by the action of ACE with the sample, and A_blank_ is the measure of unreacted HHL.

To calculate the IC_50_, (the peptide concentration required for inhibit 50% of ACE activity), dilutions were prepared at different concentrations of the hydrolysate and the percentage inhibition was calculated for each concentration tested. Subsequently, the percentage of ACE-I activity versus the concentration of hydrolysate used (µg peptides/mL) was plotted. The equation of the line (y = ax + b) was obtained and the concentration of the hydrolysate giving an inhibition activity of 50%, i.e., IC_50_ = (50 − b)/a, was calculated.

### 2.4. Peptide Identification

The identification was determined using tandem mass spectroscopy (MS) analysis using non-liquid chromatography from the NPN fraction obtained in Section 2.1. The methodology used was described by Bueno-Gavilá et al. (2019). The identification of the peptide sequences of the hydrolysates of the different hams was conducted at the Proteomics and Bioinformatics Unit of the University of Córdoba, Spain. MS2 spectra were found with SEQUEST HT against the UnitProtKB database.

The peptides of each hydrolysate were identified and quantified using the peptide spectral matches (PSM). Quantification values were normalized, focusing on the total PSM for all peptides in the sample. Thus, the quantification of a single peptide was comparable between those of the different samples. Additionally, we performed a search for each of the identified peptides in the BIOPEP-UWM database [40]. Two types of searches were performed: identification of activated biopeptides in the sample and identification of potential biopeptides containing fragments of bioactive sequences in their primary structure. Data analysis was performed with R (version 3.4.1; https://www.r-project.org, accessed on 17 December 2022).

### 2.5. Statistical Analysis

All analyses of our samples were performed in triplicate. Statistical analysis of our samples was performed using SPSS software (version 21.0, IBM Corporation, Armonk, NY, USA). To evaluate whether salt reduction and deboning affected the different analyses, a one-way ANOVA between RIB and TIB hams was performed. The effect of breed was also evaluated using a one-way ANOVA between RIB and RWC hams. When the effect of transformation or breed was significant (*p* < 0.05), the results were compared using a Fisher’s LSD test.

## 3. Results and Discussion

### 3.1. Evaluation of Antihypertensive Activity (ACE-I) in Hams with Different Curing Losses (33% and 38%)

Angiotensin-I-converting enzyme (ACE-I) is one of the key enzymes in the regulation of blood pressure, given its participation in the renin angiotensin–aldosterone system (RAAS) [41]. Figure 2 indicates the evolution of ACE inhibitory activity throughout the assay as a function of peptide concentration in hams with different curing loss.

Table 1 and Table 2 indicate the effect of pig genetic line and processing on ACE inhibitory activity, represented as the concentration of peptides necessary (mg/mL) to inhibit 50% of this activity (IC_50_). All samples showed ACE inhibitory activity, which increased with increasing concentration of the peptides. This could be due to the presence of small peptides smaller than 3 kDa [15,42,43]. The ham that showed the highest ACE-I activity (IC_50_) was RIB_33_, having a greater potential to control diseases associated with the cardiovascular system [44].

The IC_50_ in the final product was lower (higher activity) in Iberian hams than in white hams (Table 1), probably due to the longer curing time used in Iberian hams, consistent with what has been observed in other studies [15]. However, when the weight loss is 33%, no significant differences were observed between genetic lines (Table 2).

The processing method did not significantly influence ACE-I activity in Iberian hams (*p* ≥ 0.05), although it was slightly higher in salt-reduced hams (RIB_38_) (Table 1).

A study conducted at the Catholic University of Murcia (UCAM) showed that the consumption of cured ham rich in bioactive peptides has a positive influence on the regulation of glycaemia and cholesterolemia in healthy patients, so that far from being a restricted food, its regular consumption has a positive effect on modifiable risk factors associated with premature cardiovascular disease [20].

Table 3 presents the results of the effect of processing time on the production of ACE inhibitory activity. In RWC, hams with a weight loss of 33% have greater antihypertensive activity than those with a 38% weight loss (*p* ≤ 0.05). However, in Iberian hams, processing time does not imply greater ACE-I activity. In Serrano and Panxian hams, some have observed that this activity increases significantly in the last curing phase [30,42]. Furthermore, other authors have also observed this behavior for dipeptide AA, which increases its activity by 40% from 6 months to 12 months of ham curing [45]. Because ACE-I has been detected in the hams studied, it could counteract the harmful effects of sodium in the body [46].

### 3.2. Antioxidant Activity

The DPPH radical study to evaluate the antioxidant activity of samples has been described as a suitable procedure for this purpose [47].

Cured ham has been identified as a source of peptides with antioxidant activity [48]. Despite this, no studies have evaluated antioxidant activity in salt-reduced hams. DPPH scavenging activity is also a commonly used technique to evaluate antioxidant capacity. This activity is directly associated with hydrophobic AA in peptides, so these AA will exist in antioxidant peptides [49,50].

Figure 3 indicates the evolution of in vitro antioxidant activity as a function of the peptide concentration of the hams with different processing; all samples show higher antioxidant activity as the concentration of peptides increases. RWC_38_ has higher antioxidant activity, reaching 75% inhibition. In RIB_38_, we also observed an increase in antioxidant activity as the curing process progressed, higher than the healing process, higher than in RIB_33_. TIB_38_ shows lower antioxidant activity than RWC_38_ and is like RIB_38_. The ham with the lowest antioxidant activity was RIB_33_ in all the peptide concentrations we studied.

Table 4 and Table 5 indicate the in vitro DPPH radical (antioxidant) scavenging activity of the ham in the drying and final phases, respectively. The concentration (mg/mL) of each NPN needed to inhibit 50% of the antioxidant activity (IC_50_) was evaluated. All the samples studied showed antioxidant activity both in the drying phase and in the final product.

Table 4 indicates the IC_50_ values obtained for each sample and the effect of pig genetic line and processing on the antioxidant activity of the TIB, RIB, and RWC hams. Genetic line significantly influenced the uptake of the DPPH radical in these samples (*p* ≤ 0.05), as did RIB_33_ and RWC_33_ (Table 5). However, salt reduction and deboning did not influence the antioxidant activity of the samples, although it was higher in TIB_38_. The RWC_38_ hams have the highest antioxidant activity because they reached the IC_50_ with a lower peptide concentration (0.155 ± 0.013 mg/mL). These data coincide with the higher proteolysis index obtained in white hams in a previous study [3] due to the higher activity of cathepsins and calpains of this breed [4]. In Serrano hams, peptides have been identified with an IC_50_ at a concentration of 1.5 mg/mL [46]. Furthermore, Jinhua hams in eastern China, managed an IC_50_ at a lower concentration of 1 mg/mL [51]. However, in subsequent studies, Jinhua hams achieved an IC_50_ at 2.5 mg/mL, whereas Xuanwei hams required a concentration of 4.5 mg/mL [52]. In contrast to this study, others have shown that meat from purebred and Duroc-crossed Iberian pigs would be less predisposed to oxidation than those from white pig breeds [53]. Others claim that meat products such as Iberian ham have a greater antioxidant capacity than fresh ham products before being cured, or other foods such as red wine [54].

Table 6 shows that, for both salt-reduced Iberian and white hams, the increase in curing time significantly affects the antioxidant capacity of the samples (*p* ≤ 0.05), being higher in RIB_38_ and RWC_38_. This could be due to the increase observed in proteolytic activity in the later stages of curing [3], often related to the increase in temperature [23].

The results show that the antioxidant capacity of the hams increases as the curing process progresses and is not affected by the reduction in the Iberian ham. Therefore, cured hams would be a good source of antioxidant activity despite containing pro-oxidant agents such as salt and heme and even reactive oxygen species (ROS), which can cause cell damage [55,56].

### 3.3. Bioactive Peptide Sequencing

The peptides present in the samples of the hams from the five batches studied (RIB_38_, RIB_33_, RWC_38_, RWC_33_, and TIB_38_) were sequenced by LC-MS/MS analysis. Table 7 indicates the number of sequenced peptides per sample. The ham with the highest number of sequenced peptides was RWC_38_, RIB_33_ had the lowest number of peptides sequenced, and RIB_38_ presented a greater number of peptides than TIB_38_. This coincides with the values of non-protein nitrogen and the proteolysis index (PI) obtained in a previous study in hams with a loss of 38%, where the highest and lowest NPN and PI were found in salt-reduced white hams (RWC_38_) and traditionally cured hams (TIB_38_), respectively [3].

In this study, no peptides already obtained from the database were found among the peptides obtained in the proteomic study. Therefore, their bioactivity has not been demonstrated in previous studies.

In other studies, identical sequences were found in cured ham, for example, KAAAAP, AAPLAP, and KPVAAP, with origin in different types of myosin protein, were identified as the peptides with the highest ACE-I activity in Teruel PDO ham [30], and are also present in Serrano ham [20].

Their stability and their retention of bioactivity during processing and after in vitro digestion were examined. In vivo studies showed that the AAATP peptide had the highest antihypertensive activity, lowering systolic blood pressure with a short-term effect [46]. Furthermore, other sequences with antihypertensive activity were identified, such as ASGPINFT and DVITGA (both also derived from myosin protein). In another study, AAATP with the KA dipeptide had DPP4 inhibitory activity that would contribute to improving the concentration of glucose in the blood [20].

The antioxidant power is another bioactivity studied in traditional Serrano ham [46]. The SAGNPN peptide has been identified to have the greatest capacity to donate electrons, neutralizing the oxidative capacity, even more than the peptides synthesized [46,57]; furthermore, the peptide GLAGA had the highest reducing power [58]. Moreover, SNAAC and AEEEYPDL, identified in the cured ham, had high antioxidant activity [59].

Numerous bioactive peptides with a high antihypertensive power have been identified in Iberian ham, which are higher than those in Serrano ham. The sequences that are repeated most frequently, which coincide with the BIOPEP database, are PPK, PAP, and AAP [60]. However, the following dipeptides, such as EA, with ACE-I activity, or PP and VE, which showed ACE- and DPP4-inhibitory activity have also been sequenced [61].

Dipeptides with anti-inflammatory and cardiovascular protective activity (PA, GA, VG, EE, ES, DA, and DG) have been identified in hams with reduced salt content, besides contributing to the product aroma and flavor [62]. However, no studies have been found in fresh deboned and salt-reduced Iberian or white ham.

#### Study of Putative Activity Peptide Sequences

A search has been conducted for peptide precursors that may contain biopeptides in their sequence and could theoretically be activated after digestion. This technique is useful for very small sequences (less than seven Amino Acids) and by using the proteomics procedure, it is impossible to detect them.

To contextualize the type of bioactivity of the samples, a Z-scoring was performed to plot the variation between samples regarding the mean of the different activities (heatmap). The results are shown in Figure 4; a higher intensity red color means that this activity will be over-represented regarding the mean of the five samples. Bioactivities are grouped according to the intensity of occurrence in each sample. In addition, the succession of rows and columns is rearranged to avoid intersection of the dendrogram lines. Blue lines represent the value of the coefficient. Individually, we have represented in which sample each group of activities stands out for each group or clusters (Figure 5), each corresponding to a group of bioactivities. In RWC_33_, the main activity is immunostimulatory. No studies have been found on the presentation of this bioactivity in cured ham.

RWC_38_ showed the highest antioxidant activity. These results coincide with the activity observed in vitro using DPPH (Table 4). Other bioactivities that stand out in this sample are those of neuropeptide activation, hypolipemic, anti-inflammatory, anti-cancer, and hypotensive activities. Antioxidant and hypotensive activity have also been well studied in white pig hams.

Several studies confirm the occurrence of these bioactivities in ham [48]. Recently, peptides with anti-inflammatory activity have been identified in Xuanwei hams, showing reduced symptoms of inflammatory bowel disease in mice, and it has been pro-posed that these peptides could be a functional drug in patients suffering from this disease [63].

In RIB_33_ hams, the predominant activities are stimulatory, immunomodulatory, a CaMPDE inhibitor, a DPP4 inhibitor, antithrombotic, and ACE-I, consistent with our results for antihypertensive activity (Table 2), where RIB_33_ had the highest ACE-I. Likewise, the Iberian ham showed greater ACE-I activity compared to the traditional Serrano hams [60].

Antihypertensive activity is well studied in ham [20,21,60]. There are studies that would claim that Serrano ham would be a good source of DPP4 and that these peptides could be an adjunct in the treatment of type 2 diabetes [64].

RIB_38_ hams have HMG-CoA reductase inhibitory, regulatory, and immunological activity. HMG-CoA reductase inhibitors play an important role in the control of hyper-cholesterolemia and, indirectly, in the control of the onset of cardiovascular disease. Other studies have found dipeptides such as DA, DD, EE, ES, and LL in cured ham, which have been identified as the main inhibitors of this coenzyme [65]. Furthermore, TIB_38_ hams stand out for their binding, ubiquitin mediator protein activator, renin inhibitor, dipeptidyl peptidase III inhibitor, and embryotoxic activity, bioactivities that have not yet been studied.

Each group of bioactivities was represented by a color (Figure 4). In Figure 5, we can observe six clusters, one for each group of bioactivities, where the values of that group of bioactivities are quantified for each sample.

Because bioactive sequence fragments are found in the samples, a spider web plot with normalized quantification of the peptide precursors of the five hams is shown in Figure 6. This distribution allows differentiation between hams according to activity. The potential bioactivity of the peptides identified in each sample is reflected by using the same scale and amplitude and the same scale and width of the axis, allowing comparison between them.

Cured ham is considered a good source of different bioactive peptides that have important functional activities, such as the inhibition of the angiotensin converting enzyme, hypoglycemic, and anti-inflammatory activities [29].

### 3.4. Bioactivity Analysis Based on Amino Acid Composition

The composition of Amino Acid (AA) used to analyze the bioactivity of the samples was conducted on 38% cured hams, as these had the best organoleptic characteristics and, therefore, would be destined for the end consumer.

In bioactivity studies, it is important to consider the structural properties of sequences [66]. Certain characteristics, such as size, hydrophobicity, and composition, may influence the stability or bioavailability of the peptides. Approximately 20 sequences were selected from those identified in each ham with less than 1.5 kDa and with a maximum of 12 AA in their chain. Processing time causes the size of the peptides to decrease and increases the antioxidant activity of the peptides [49], as short AA sequences are more likely to be bioactive [62,67]. In addition, over 50% of the AAs in the chain should be hydrophobic, as this contributes to antioxidant activity [68]. The presence of AAs A, D, E, G, L, P, and V confers antioxidant and antihypertensive activity on the peptide sequence [68,69,70], and this activity is directly related to the molecular weight of the peptide sequence [71]. However, the presence of H, Y, W, F, M, and C could inhibit free radicals by direct electron transfer [67]. The amino acid sequences of the peptides identified from salt-reduced Iberian hams (RIB) are shown in Table 7.

### 3.5. Identification of Peptides Present in RIB Hams

The AA sequences of the peptides identified from the hydrolysates of salt-reduced Iberian hams (RIB) are shown in Table 8.

Antioxidant activity is highly present among the selected sequences. Some have over 50% of the peptides that provide antioxidant activity. The LDLALEKD, AAFPPDVGGN, AGNPDLVLPV, and AFGPGLEGGL peptides stand out for having over 80% of AAs that would favor antioxidant activity, with AFGPGLEGGL having the highest antioxidant activity (90% of its AAs).

The AAFPPDVGGN and AFPPDVGGN have been identified as present in pork [72] and six sequences containing them have been found (AFPPDVGGN, AAFPPDVGGN, AFPPDVGGNV, AAFPPDVGGGGNV, AFPPDVGGGGNVD, and AAFPPDVGGGGNVD). The peptides FPPDVGGN and FPPDVGGNVD originating from the protein could also be derived from these sequences, identified as myosin [46]. From the action of the enzyme, dipeptidyl peptidase [73] could be released from some sequences as the VD dipeptide, which would have DPP4 inhibitory activity and, therefore, anti-diabetic activity [64,74].

The most prominent sequence is CLFVCR, as it has 83% of hydrophobic AAs, 67% of AAs conferring ACE-I activity, and 50% of AAs scavenging free radicals. ACE-I activity would be more present in sequences containing hydrophobic AA residues in the three C-terminal positions [75]. For this sample, the sequence AGNPDLVLPV has three hydrophobic AAs at the C-terminus. The dipeptide WK could be extracted from longer peptides originating from β-enolase, such as DGADFAKW (Table 8). This dipeptide has been identified as an inhibitor of DPP4 [76,77]. Likewise, the sequences LIGIEVPH, IDLIEKPM, FDKIEDMA, WNDEIAPQ, and DLDISAPQ originate from the IE and SI dipeptides of the α-enolase protein; they have been described as ACE- and DPP4-inhibitory peptides, respectively [74,78]. These dipeptides could be responsible for the high antihypertensive activity observed in this study for sample RIB_38_ (Table 1).

Recently, some dipeptides related to anti-inflammatory activity, which could confer cardiovascular protection, have been identified in salt-reduced cured hams [62]. These dipeptides are PA, GA, DA, and DG and could be derived from sequences found in RIB_38_ (ALQPALKF, WNDEIAPQ, MADTFLEH, DLDISAPQ, DGADFAKW, MADTFLEH, and AGNPDLVLPV), with GA being mainly identified in the study.

Table 9 indicates the prominent peptide sequences detected in the ham samples of traditionally cured Iberian ham (TIB). In these samples, six of the selected sequences presented over 80% of the AAs that could provide antioxidant activity to the product (AFPPDVGGNV, AAFPPDVGGN, DVVLPGGNL, VAVGDKVPAD, DIAVDGEPLG AGNPDLVLPV, and AFGPGLEGGL). RIB_38_ has the highest antioxidant activity. However, the sequence that stands out for having the highest amount of hydrophobic peptides is ILPGPAPW. This peptide comprises the Pro-Ala-Pro sequence, one of the most repeated sequences among the bioactive peptides described in the literature [15], which would confer good antioxidant activity to the sample [68]. Furthermore, these sequences could contribute to the bioactivity described for TIB_38_ ham (Figure 4).

Four sequences (ILPGPAPW, VMGAPGAPM, GDLGIEIPA, and AGNPDLVLPV) have three hydrophobic AAs at the C-terminus, and are therefore more likely to develop ACE-I activity [75]. The AGNPDLVLPV sequence matches that found in RIB_38_. In the GDLGIEIPA and IELIEKPM sequences, we can find the dipeptide IE dipeptide related to ACE-I bioactivity [74]. The same six sequences identified in RIB_38_ have also been found in TIB_38_ (AAFPPDVGGNV, AAFPPDVGGN, AFPPDVGGNVD, AAFPPDVGG-NVD, AFPPDVGGN, and AFPPDVGGNV), have been identified in pork, and could have inhibited DPP4 derived from the dipeptide DV [61].

From a comparison study between traditional and salt-reduced cured hams, di-peptides such as DA, PA, and VG would be present in a higher proportion in traditional cured hams [62]. The last two sequences of Table 9 could derive from the peptides found in sample TIB_38_ and could contribute to its anti-inflammatory and antihypertensive activity. Other sequences, which were identified in this study, are GA (ACE and DPP4 inhibitory activities) and DG (ACE-I activity).

The selected AA sequences of salt-reduced white hams (RWC) are shown in Table 10. The sequences that stand out for having over 80% of AAs and confer antioxidant activity are DLAEDAPW and AEVIALPVE. The latter sequence is also present in RIB_38_, with the highest antioxidant activity.

The sequence that stands out for having the highest amount of hydrophobic AAs is ILPGPAPW, the same as TIB_38_, and has one of the most repeated sequences among bio-active peptides (PAP) [15]. Furthermore, this sequence has 75% of AAs that confer antioxidant activity, three hydrophobic AAs at the C-terminus that confer ACE-I activity, and 13% of AAs that could inhibit free radicals. However, six peptide sequences with three hydrophobic AAs at the C-terminus (ILPGPAPW, AVIGPSLPL, VMGAPGAPM, ISAPSADAPM, DLAEDAPW, and GDLGIEIPA) were found in the RWC_38_ samples, which conferred ACE-I activity. However, in RIB_38_, of the sequences selected, none had over two hydrophobic AAs at the C-terminus. The sequences ILPGPAPW, VMGAP-GAPM, and GDLGIEIPA match TIB_38_. Furthermore, the LKGADPEDVITGA and GADPEDVITGA would contain the bioactive peptide DVITGA in their chain, related to high ACE-I activity due to the presence of AA alanine at the C-terminus [39,46]. Despite this, the RWC_38_ ham showed the least antihypertensive activity (Figure 4); it would be necessary to study if these peptides confer ACE-I activity and in what quantity they are present. There are more sequences from which the dipeptide IE could be derived, already described as a precursor of this bioactivity [74].

The sequence identified in RWC_38_, FKAEEEYPDLS, once digested, could cause the peptide AEEEYPDL, derived from protein creatine kinase and identified as a potent antioxidant [59]. Using multiple reaction monitoring (MRM), it was quantified at a concentration of 0.148 fg/g in cured ham [79]. This could explain why the RWC_38_ hams showed the highest antioxidant activity (Figure 4) and the highest rate of proteolysis obtained in white hams [3].

In RWC_38_ hams, the same six sequences described in RIB_38_ and TIB_38_ (AAFPPDVGGNV, AAFPPDVGGNV, AFPPDVGGNVD, AAFPPDVGGNVD, AFPPDVGGNV, and AFPPDVGGNV) have been identified. However, this would explain the difference in proteolytic activity [3] between different pig genetic lines and influence of processing, because salt-reduced Iberian hams (RIB_38_ and RIB_33_) had the highest DPP4 inhibitory activity, lower than white hams (RWC_38_ and RWC_33_) and traditional Iberian ham (TIB_38_).

In a recent study of salt-reduced white ham, hydrophobic PA dipeptides (related to bitter taste and with ACE-I and anti-inflammatory activity) and VG (related to bitter and umami taste and with ACE-I activity) were identified that could be derived from the sequences identified in our sample [62].

Different unique and common sequences that could act as peptide precursors and that have been identified in the samples would be responsible for the bioactivities found in the different types of ham. The results show that in RIB_38_ ham, the precursors found could be responsible for its high antihypertensive capacity, noting that the change in processing varies the sequences identified in both samples (RIB_38_ and TIB_38_). Furthermore, peptides already referenced in the literature have been found in the RWC_38_ ham, including a sequence that gives rise to a potent antioxidant peptide (AEEEYPDL) that would explain its increased bioactivity. However, none of these co-inciding peptides are found in Iberian hams.

## 4. Conclusions

Salt-reduced boneless hams presented a higher concentration of peptides and higher bioactivity compared to traditionally cured hams, due to higher proteolysis.

The salt-reduced white ham had the highest antioxidant activity and the salt-reduced Iberian ham the highest antihypertensive activity. Antioxidant activity was significantly influenced by pig genetic line and antihypertensive activity was modified by pig genetic line in the final product. In addition, salt-reduced white ham had the greatest hypolipidemic, anti-inflammatory, and anticarcinogenic activity. However, salt-reduced Iberian ham stood out for its significant HMG-CoA reductase inhibitory, regulatory, and immunological activity.

Salt reduction has had a positive influence on the bioactivity of the hams, with salt-reduced hams, both Iberian hams and hams from white pigs, having the highest bioactivity compared to traditionally cured Iberian hams.

In all types of hams, peptide precursors sequenced could give rise to sequences identified as bioactive in literature, with white pig hams showing the highest quantity. The dipeptide DV is present among the precursors of all hams and the bioactive peptides DVITGA and AEEEYPDL in the precursors of the reduced white hams.

## Figures and Tables

**Figure 1 foods-12-01022-f001:**
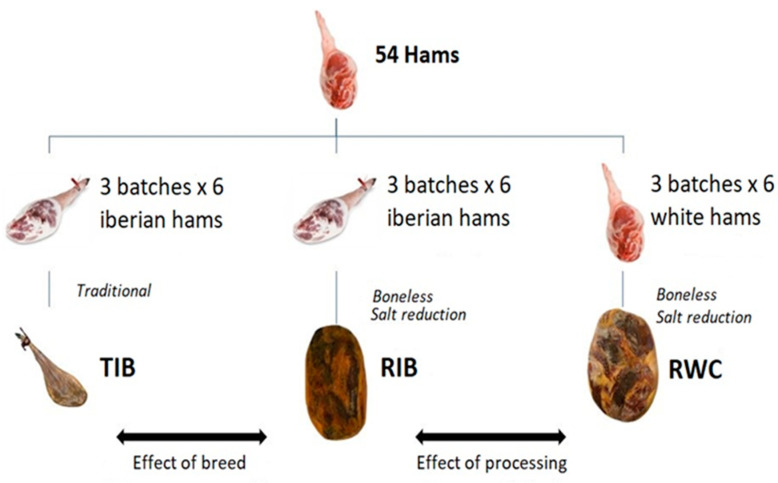
Outline of the experimental design.

**Figure 2 foods-12-01022-f002:**
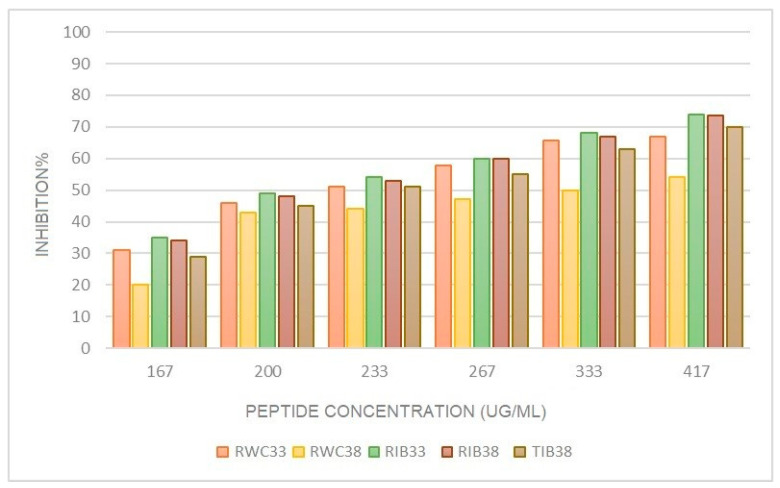
Evolution of ACE-I activity as a function of peptide concentration in hams with 33% and 38% curing loss.

**Figure 3 foods-12-01022-f003:**
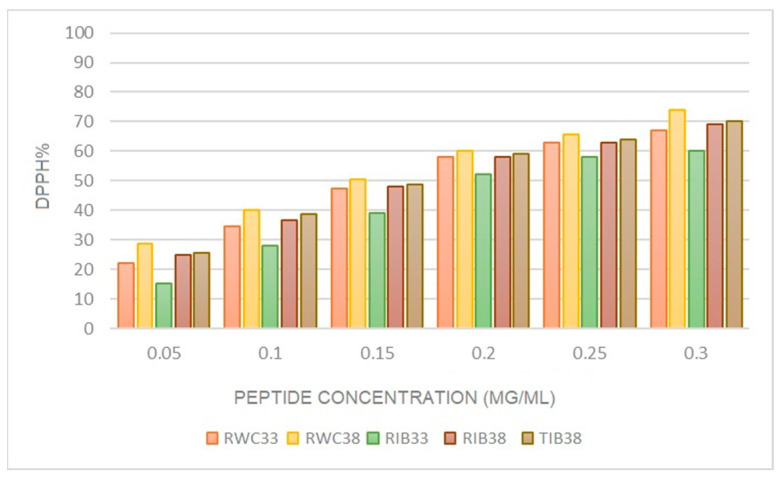
Evolution of in vitro antioxidant activity as a function of peptide concentration in hams with 33% and 38% curing loss.

**Figure 4 foods-12-01022-f004:**
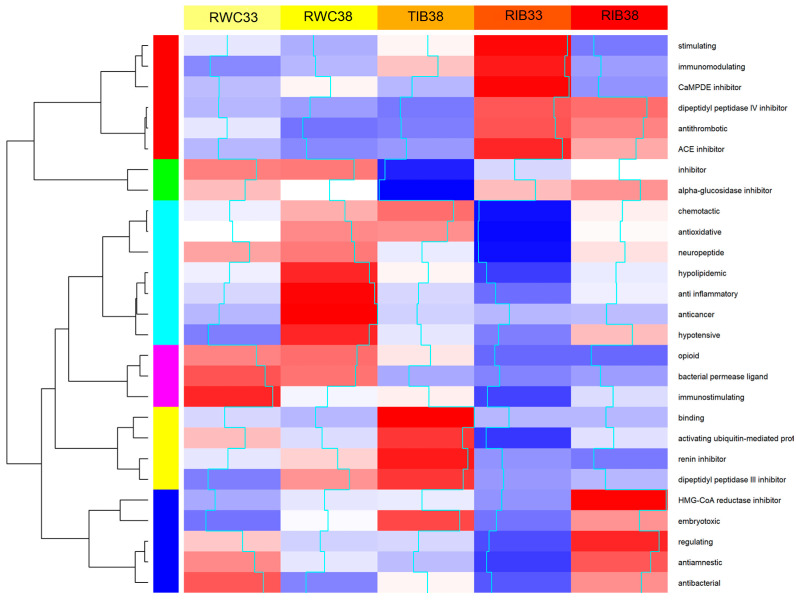
Heatmap and dendrogram of bioactivities of the different ham samples studied. Quantification of bioactivity is regarding the mean. The grouping relationship between the groups of activities is defined.

**Figure 5 foods-12-01022-f005:**
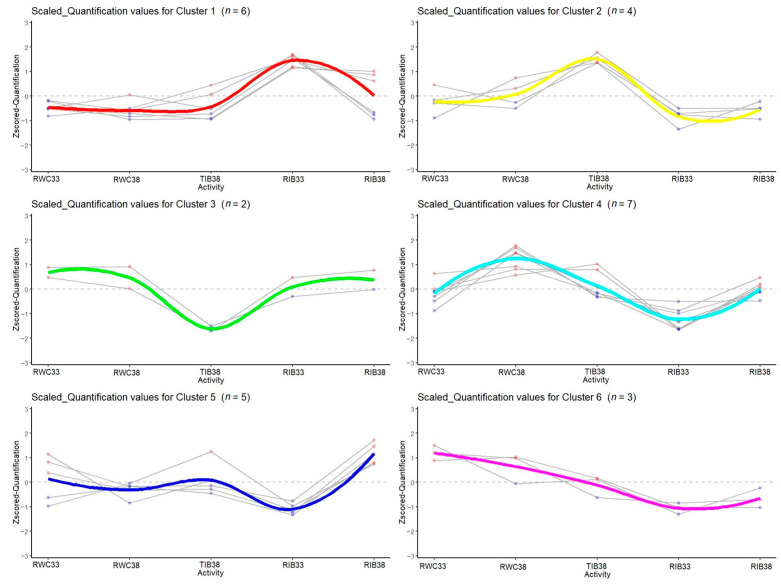
Representation of the bioactivity in each sample for each cluster of activities.

**Figure 6 foods-12-01022-f006:**
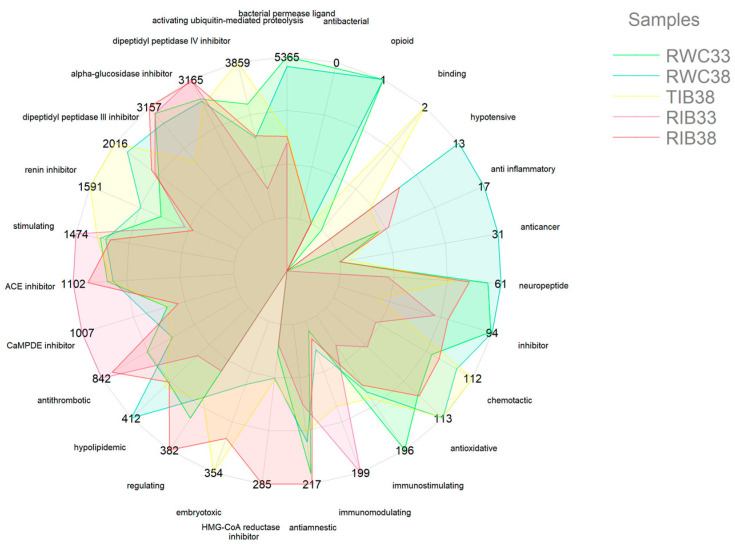
Standardized quantification of peptide precursors (×103) in different cured ham samples.

**Table 1 foods-12-01022-t001:** Effect of pig genetic line and processing on ACE-I activity in hams with 38% cure loss.

	Dry-Cured Ham Type			*p*-Value	
Formulation	TIB	RIB	RWC	Genetic Line	Processed
IECA (IC_50_)	0.249 ± 0.018 ^b^	0.220 ± 0.012 ^b^	0.335 ± 0.017 ^a^	0.025	0.387

One-way ANOVA. ^a,b^ Values within a row with different superscripts differ significantly at *p* ≤ 0.05 (Fisher LSD Test). TIB: traditional Iberian dry-cured ham; RIB: reduced Iberian dry-cured ham; RWC: reduced white pig commercial crosses. ACE inhibitor (IC_50_): peptide concentration required (mg/mL) to inhibit 50% of ACE activity. *p*-Value production line: one-way ANOVA RIB vs. RWC; *p*-value processed: one-way ANOVA TIB vs. RIB (*p*-value significant at *p* ≤ 0.05). SEM: standard error of the mean.

**Table 2 foods-12-01022-t002:** Effect of pig genetic line on ACE-I activity in hams with a 33% curing loss.

	Dry-Cured Ham Type		*p*-Value
Formulation	RIB	RWC	Genetic Line
IECA (IC_50_)	0.215 ± 0.043 ^a^	0.230 ± 0.014 ^a^	0.7550

One-way ANOVA. ^a^ Values within a row with different superscripts differ significantly at *p* ≤ 0.05 (Fisher LSD Test). RIB: reduced Iberian dry-cured ham; RWC: reduced white pig commercial crosses dry-cured ham. ACE inhibitor (IC_50_): peptide concentration required (mg/mL) to inhibit 50% of ACE activity. *p*-value production line: One-way ANOVA between RIB and RWC (*p*-value significant at *p* ≤ 0.05). SEM: standard error of the mean.

**Table 3 foods-12-01022-t003:** Effect of the decrease in ACE-I activity of salt-reduced Iberian and white hams (RIB and RWC).

Dry-Cured Ham Type			
		RIB	RWC
IECA (IC_50_)	33% reduction	0.215 ± 0.043 ^a^	0.230 ± 0.014 ^a^
	38% reduction	0.220 ± 0.012 ^b^	0.335 ± 0.017 ^a^
*p*-value	reduction	0.486	0.001

One-way ANOVA. ^a,b^ Values within a row with different superscripts differ significantly at *p* ≤ 0.05 (Fisher LSD Test). RIB: reduced Iberian dry-cured ham; RWC: reduced white pig commercial crosses dry-cured ham. ACE inhibitor (IC_50_): peptide concentration required (mg/mL) to inhibit 50% of ACE activity. *p*-Value reduction: one-way ANOVA 38% reduction vs. 33% reduction. The results are expressed in mg/mL as mean ± SEM. SEM: standard error of the mean.

**Table 4 foods-12-01022-t004:** Effect of pig genetic line and processing on antioxidant activity in hams with a curing loss of 38%.

Dry-Cured Ham Type				*p*-Value	
Formulation	TIB	RIB	RWC	Processed	Genetic Line
DPPH (IC_50_)	0.173 ± 0.054 ^a^	0.199 ± 0.048 ^a^	0.155 ± 0.013 ^a^	0.455	0.048

One-way ANOVA. ^a^ Values within a row with different superscripts differ significantly at *p* ≤ 0.05 (Fisher LSD Test). TIB: traditional Iberian dry-cured ham; RIB: reduced Iberian dry-cured ham; RWC: reduced white pig commercial crosses. DPPH (IC_50_): peptide concentration required (mg/mL) to inhibit 50% of the antioxidant activity. *p*-value production line: One-way ANOVA between TIB, RIB, and RWC (*p*-value significant at *p* ≤ 0.05). SEM: standard error of the mean.

**Table 5 foods-12-01022-t005:** Effect of pig genetic line on the antioxidant activity of hams with a 33% curing loss.

	Dry-Cured Ham Type		*p*-Value
Formulation	RIB	RWC	Genetic Line
DPPH (IC_50_)	1.888 ± 0.041 ^b^	0.197 ± 0.013 ^a^	0.001

One-way ANOVA. ^a,b^ Values within a row with different superscripts differ significantly at *p* ≤ 0.05 (Fisher LSD Test). RIB: reduced Iberian dry-cured ham; RWC: reduced white pig commercial crosses dry-cured ham. DPPH (IC_50_): peptide concentration required (mg/mL) to inhibit 50% of the antioxidant activity. *p*-value production line: One-way ANOVA between RIB and RWC (*p*-value significant at *p* ≤ 0.05). SEM: standard error of the mean.

**Table 6 foods-12-01022-t006:** Effect of reduction in the DPPH activity of salt-reduced Iberian and white hams (RIB and RWC).

Dry-Cured Ham Type			
		RIB	RWC
DPPH (IC_50_)	33% reduction	1.888 ± 0.041 ^b^	0.197 ± 0.013 ^b^
	38% reduction	0.199 ± 0.048 ^a^	0.155 ± 0.013 ^a^
*p*-value	reduction	0.000	0.043

One-way ANOVA. ^a,b^ Values within a row with different superscripts differ significantly at *p* ≤ 0.05 (Fisher LSD Test). RIB: reduced Iberian dry-cured ham; RWC: reduced white pig commercial crosses dry-cured ham. DPPH (IC_50_): peptide concentration required (mg/mL) to inhibit 50% of the antioxidant activity. *p*-value reduction: one-way ANOVA 38% reduction vs. 33% reduction. The results are expressed in mg/mL as mean ± SEM. SEM: standard error of the mean.

**Table 7 foods-12-01022-t007:** Number of peptides sequenced per sample.

Sample	Number of Peptides Sequenced
RIB_38_	979
RIB_33_	602
RWC_38_	1053
RWC_33_	780
TIB_38_	904

**Table 8 foods-12-01022-t008:** Identification of amino acid sequences of peptides present in salt-reduced Iberian ham (RIB) using LC-MS/MS.

N	Peptide Seq	Exp. Mass	Protein Source	Acc
1	CLFVCR	750.37	Ubiquitin-associated and SH3 domain containing B	F1S9R5
2	DLDISAPQ	858.43	Calsequestrin	F1RJW7
3	AFPPDVGGN	873.41	Myosin regulatory light chain 2	Q5XLD2
4	ALQPALKF	887.53	Superoxide dismutase	A0A287A4Z2
5	LYKVAVGF	896.52	Alpha-amylase	I3LSA5/F1S574/F1S573
6	FDKPVSPL	902.5	Creatine Kinase M-type/Mitochondrial Creatine kinase 2	Q5XLD3/Q2HYU1
7	DGADFAKW	909.41	Fructose-biphosphate aldolase	A0A287A1V5
8	AAVKELATL	915.55	Carboxypeptidase B	F1SKC7
9	LDLALEKD	916.5	Alpha-amylase	I3LSA5/F1S574/F1S573
10	AFGPGLEGGL	917.47	Filamin C	F1SMN5
11	AAFPPDVGGN	944.45	Myosin regulatory light chain 2	Q5XLD2
12	DNDIMLIK	961.5	Uncharacterized Protein	A0A287B5W2
13	MADTFLEH	963.42	Pyruvate Kinase	A0A287B8G0
14	MIADYLNK	967.49	Alpha-amylase	I3LSA5/F1S574
15	FDKIEDMA	968.44	Myosin 2-4-7	F1SS64/Q9TV62/A0A2867PQ9
16	LGIDVWEH	968.48	Superoxide dismutase	A0A287A4Z2
17	WNDEIAPQ	972.44	Phosphoglycerate mutase	B5KJG2
18	AFPPDVGGNV	972.48	Myosin regulatory light chain 2	Q5XLD2
19	AGNPDLVLPV	994.56	Beta-enolase	A0A287AZR0
20	AAFPPDVGGNV	1043.52	Myosin regulatory light chain 2	Q5XLD2
21	AFPPDVGGNVD	1087.51	Myosin regulatory light chain 2	Q5XLD2
22	AAFPPDVGGNVD	1158.54	Myosin regulatory light chain 2	Q5XLD2

**Table 9 foods-12-01022-t009:** Identification of the amino acid sequences of peptides present in traditional Iberian ham (TIB) using LC-MS/MS.

N	Peptide Seq	Exp. Mass	Protein Source	Acc
1	VMGAPGAPM	830.39	Uncharacterized Protein	F1RVL5
2	ILPGPAPW	850.48	PDZ and LIM domain protein 3	Q6QGC0
3	DIDLSAPQ	858.42	Calsequestrin	F1RJW7
4	AFPPDVGGN	873.41	Myosin regulatory light chain 2	Q5XLD2
5	GDLGIEIPA	884.47	Pyruvate kinase	A0A287B8G0
6	IGIGPGGVIGA	910.54	Uncharacterized protein	A0A287AXV0
7	AFGPGLEGGL	917.47	Filamin C	F1SMN5
8	AAFPPDVGGN	944.45	Myosin regulatory light chain 2	Q5XLD2
9	TVPPAVPGIT	951.55	Fructose-biphosphate aldolase	A0A287A1V5
10	VAVGDKVPAD	970.52	Calcium-transporting ATPase	A0A287APK5
11	AFPPDVGGNV	972.48	Myosin regulatory light chain 2	Q5XLD2
12	IELIEKPM	972.54	Myosin 2–4	F1SS65/Q9TV62
13	DVVVLPGGNL	982.56	DJ-1 Protein	Q0R678
14	HMWPGDIK	983.48	Alpha-amylase	I3LSA5/F1S574
15	FNVIQPGPI	984.55	2,4-dienoyl-CoA reductase 1	D6QST6
16	DIAVDGEPLG	985.48	Peptidyl-prolyl cis-trans isomerase A	P62936
17	AGNPDLVLPV	994.56	Beta-enolase	A0A287AZR0
18	RIPADVDPL	995.55	Alpha-crystallin B chain	A0A287ATJ4
19	NGAHIPGSPF	996.49	Filamin C	F1SMN5
20	AAFPPDVGGNV	1043.52	Myosin regulatory light chain 2	Q5XLD2
21	AFPPDVGGNVD	1087.51	Myosin regulatory light chain 2	Q5XLD2
22	AAFPPDVGGNVD	1158.54	Myosin regulatory light chain 2	Q5XLD2

**Table 10 foods-12-01022-t010:** Identification of the amino acid sequences of peptides present in salt-reduced White ham (RWC) using LC-MS/MS.

N	Peptide Seq	Exp. Mass	Protein Source	Acc
1	VMGAPGAPM	829.39	Uncharacterized protein	F1RVL5
2	AKLPADTE	843.44	Triosephosphate isomerase	A0A286ZRV2
3	SGMNVARL	846.43	Pyruvate Kinase	A0A287B8G0
4	ILPGPAPW	849.48	PDZ and LIM domain protein C	Q6QGC0
5	AFPPDVGGN	873.41	Myosin regulatory light chain 2	Q5XLD2
6	LIGIEVPH	876.51	Uncharacterized Protein	F1RK48
7	IKIIAPPE	879.55	Actin Alpha skeletal muscle	P68137
8	GDLGIEIPA	883.47	Pyruvate kinase	A0A287B860
9	ALQPALKF	886.53	Superoxide dismutase	A0A287A4Z2
10	LYKVAVGF	895.52	Alpha-Amylase	I3LSA5
11	DLAEDAPW	915.40	Filamin C	F1SMN5
12	AAFPPDVGGN	944.45	Myosin regulatory light chain 2	Q5XLD2
13	LSVEAPLPK	952.57	Heat shock protein beta-1	A0A2C9F366
14	ISAPSADAPM	958.45	Glyceraldehyde-3-phosphate dehydrogenase	A0A287BG23
15	IELIEKPM	971.54	Myosin-2	F1SS65
16	AFPPDVGGNV	972.48	Myosin regulatory light chain 2	Q5XLD2
17	AEVIALPVE	981.55	Cut a divalent cation tolerance homolog	F1RZR6
18	LVIIEGDLE	999.56	Tropomyosin alpha 3-chain	A0A287AID2
19	AAFPPDVGGNV	1043.52	Myosin regulatory light chain 2	Q5XLD2
20	GADPEDVITGA	1044.48	Myosin regulatory light chain 2	Q5XLD2
21	AFPPDVGGNVD	1087.51	Myosin regulatory light chain 2	Q5XLD2
22	AAFPPDVGGNVD	1158.54	Myosin regulatory light chain 2	Q5XLD2
23	LKGADPEDVITGA	1285.66	Myosin regulatory light chain 2	Q5XLD2
24	FKAEEEYPDLS	1327.61	Creatine Kinase M-type	Q5XLD3

## Data Availability

Data is contained within the article.

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
