# Peer review of "Influence of Pig Genetic Line and Salt Reduction on Peptide Production and Bioactivity of Dry-Cured Hams"

_foods, 2023, doi:10.3390/foods12051022_

Round 1

Reviewer 1 Report

General comment

The topic faced in the paper is very important as regards the evaluation of the nutritional and biochemical profile of dry cured ham. The goal of demonstrating how the lower salt content and the genetic line of pigs used can influence its peptide profile is ambitious. In any case, it would be important to better clarify the experimental design by specifying, for example, if the replicas were made at different times (influence of season?). Furthermore, in my opinion, there is a lack of information regarding the experimental design (i.e. ingredients used and amounts) and the important measurements on the final product that define its composition (salt%, proteolysis index, etc.). Major revisions are needed in the “material and methods” sections as reported in the specific comments. Minor revisions are required to correct tables and figures.

Title: Please, revise the title. In my opinion, the term “production” is not suitable or associated with pig genetic lines or with peptides that are generated during processing. I would suggest you not to specify “boneless ham” since traditional control hams are also listed.

Abstract

Please, better explain the link between consumption of processed meat products and cardiovascular disease. Salt is primarily responsible for this.

Line 21- Please, choose an alternative for “production”.

Line 23-26- Please, check and correctly reformulate the final sentence.

Introduction

Line 51- Please, check the sentence “because of this naturation”.

Line 53- please, check the sentence “the final aroma of the process”.

Line 65-66- The temperature is not a parameter influenced by salt, but it affects proteolysis. Please, explain better.

Line 68-69. The ultra-processed products do not include dry cured ham in which salt and nitrites/nitrates are added. Please, I would review this reference and the related comment.

Materials and Methods

It is important to better explain the experimental design in term of batches (Do you perform the trials in the same time?) and the sampling method (biceps femoris for all hams and analysis? how the samples were stored before analysis).

Line 112-116- Please, better explain the term you used (shrinkage, curing loss, weight loss or reduction?).

Line 130-137- Reference should be given to support the method used.

Line 145-146- Please, better explain the preparation of calibration curve with Trolox.

Line 153- Please, explain the term “RSA”.

Line 158-164- Please, better explain the ACE method. Insert the reference number and specify how the results of this activity are expressed.

Line 181-182. Please, better explain the statistical analysis you performed (software and procedure with its aim). The term “triplicate” is referred to data measurements?

Results and discussion

Section 3.1- Please better explain the figure 2 (the figure is not in English) and specify which aspect you want to highlight. Figure 3 is not in English.

Section 3.1 and 3.2. Please, in this section, in all the tables related specify the measurement units of DPPH and ACE activities.

Line 200- Please, review this comment related to ham.

Line 300- Please, better explain the role of oxygen as pro-oxidant in dry cured ham.

Table 9 and table 10. Please, verify the correct reference of results (RWC or TIB).

 Conclusions

Line 527-530. Please, better explain how this comment can be extrapolated from the data presented in this research article.

Concerning the conclusion, once some aspects of the experimental design have been clarified, I could revise them better.

Reviewer 2 Report

The manuscript (foods-2211309) evaluated the influence of pig production line and salt reduction on peptide production and bioactivity of boneless hams. This is an interesting paper. Suggestions are provided below:

1. Line 34: should be “prevalent, and these foods” rather “prevalent These foods”.

2. Line 42-52: It is suggested that the author make a proper review of the research progress of prosciutto derived peptides, so as to better understand the research topic.

3. Line 54-558: It is suggested that the author add some new literatures on peptides

[1] Xiang, Z.; Xue, Q.; Gao, P.; Yu, H.; Wu, M.; Zhao, Z.; Li, Y.; Wang, S.; Zhang, J.; Dai, L. Antioxidant peptides from edible aquatic animals: Preparation method, mechanism of action, and structure-activity relationships. Food Chem. 2023, 404(Pt B), 134701.

[2] Qiao, Q.Q.; Luo, Q.B.; Suo, S.K.; Zhao, Y.Q.; Chi, C.F.; Wang, B. Preparation, characterization, and cytoprotective effects on HUVECs of fourteen novel angiotensin-I-converting enzyme inhibitory peptides from protein hydrolysate of tuna processing by-products. Front. Nutr. 2022, 9, 868681.

[3] Tyagi, A.; Chelliah, R.; Banan-Mwine Daliri, E.; Sultan, G.; Madar I.H.; Kim, N.; Shabbir, U.; Oh, D.H. Antioxidant activities of novel peptides from Limosilactobacillus reuteri fermented brown rice: A combined in vitro and in silico study. Food Chem. 2023, 404(Pt B), 134747.

4. Line 65: should be “proteolysis are parameters” rather “proteolysis, are parameters”

5. Line 75-78: This sentence is too vague, and please rewrite this sentence.

6. Line 93-95: Please rewrite this sentence “no studies have been conducted on the bioactivity of peptides in salt-reduced hams, and no studies have analyzed the effect that salt reduction would have on the ham.”

7. Line 135: should be “flask. After filtration” rather “flask; after filtration”.

8. Figure 2 and 3: It is suggested that the author change the abscissa and ordinate into English to make it easier for readers to read.

9. In the manuscript, it is suggested that the author change the ACE activity to ACE inhibitory activity, which is consistent with most ACE inhibitory peptide papers and is easier for readers to read.

10. In addition, it is suggested that the author unify ACE and ACE-I. Because ACE-I is the abbreviation for ACE inhibitory in most papers.

11. In Figure 3, the ordinate is DPPH or %DPPH RSA (%) ?

Reviewer 3 Report

The manuscript was aimed to investigate the effects of different pig production lines and different salt application on the production and bioactive properties of ham. The experiment was well designed and more importantly the results has been thoroughly analyzed and discussed by the authors. The study could bring some advancements to understand the bioactive peptides of hams as influenced by different factors.

Line 17: Changing is to was.

Line 34: A period is missed after the word of prevalent.

Line 42: Peptides are inactive precursor protein? Is this correct?

Line 52-53: Could the authors provide more explanations about why and how peptides could directly modify the meat texture? We all well know that peptides could contribute to the aroma of cured ham.

Line 57-58: Could the best results and evidence stand? It is an obscure statement.

Line 65-72: Any reports supporting that consumption of cured ham increased these healthy problems?

Line 98: Changing is to was.

Line 119: I guess these salts are from some companies?

Line 119-129: Instead of temperature, the readers may be more interested in the processing days for different stages.

Line 182-183: Clearly showing which data are used this analysis method? How about other data?

Figure 2, 3: The words in the Figures should use English instead of Spanish!

Table 1, 2, 3, 4, 5, 6: The letters in the table should be superscript instead of subscript.

Line 222: Deleting the - for the word fac-tors.

Line 331-332: Maybe some references are provided here.

Line 406: Amino acid composition?

Round 2

Reviewer 2 Report

The authors have well answered all of my concerns. Therefore, I think the manuscript (foods-2211309) could be accepted and published in Foods.